# Design of Neural Network Quantizers for Networked Control Systems

**Juan Esteban Rodriguez Ramirez** [1]  **and Yuki Minami** [2,*]

1   Graduate School of Information Science, Nara Institute of Science and Technology, 8916-5 Takayamacho, Ikoma, Nara 630-0192, Japan; juan.rodriguez.jj4@is.naist.jp
2   Department of Mechanical Engineering, Graduate School of Engineering, Osaka University, 2-1 Yamadaoka, Suita, Osaka 565-0871, Japan
*   Correspondence: minami@mech.eng.osaka-u.ac.jp

**Abstract:** Nowadays, networked control systems (NCSs) are being widely implemented in many applications. However, several problems negatively affect and compromise the design of practical NCSs. One of them is the performance degradation of the system due to quantization. This paper aims to develop dynamic quantizers for NCSs and their design methods that alleviate the effects of the quantization problem. In this paper, we propose a type of dynamic quantizers implemented with neural networks and memories, which can be tuned by a time series data of the plant inputs and outputs. Since the proposed quantizer can be designed without the model information of the system, the quantizer could be applied to any system with uncertainty or nonlinearity. This paper gives two types of quantizers, and they differ from each other in the neural networks structure. The effectiveness of these quantizers and their design method are verified using numerical examples. Besides, their performances are compared among each other using statistical analysis tools.

**Keywords:** networked control systems; quantizer; neural networks; model-free design

## 1. Introduction

The networked control systems (NCSs) are systems in which its elements are physically separated but connected by some communication channels. They have been around for some decades already, and they have been implemented successfully in many fields such as industrial automation, robotics, and power grids. Although the NCSs provide several advantages to the systems, it is well-known that one of the problems is the system performance degradation caused by the data rate constraints in the communication channels [1–3]. In the case that operation signals of NCSs are transmitted over networks under data rate constraints, the signal quantization is a fundamental process in which a continuous-valued signal is transformed into a discrete-valued one. However, the quantization error between the continuous-valued signal and discrete-valued one occurs, and it affects the performance of the NCSs. Therefore, one of the significant works is to develop a method to minimize the influence of the quantization error to the performance of the NCSs.

It has been proven that properly designed feedback-type dynamic quantizers are effective to reduce this degradation in the system's performance [4]. Several studies have considered the design of dynamic quantizers. For instance, a mathematical expression of an optimal dynamic quantizer for time-invariant linear plants was presented in [4], and an equivalent expression for nonlinear plants was introduced in [5]. Furthermore, design methods for dynamic quantizers that minimize the system's performance degradation and satisfy the channel's data rate constraints were developed in [6–8]. Then, in [9] event-triggered dynamic quantizers that reduce the traffic in the communication network were proposed. In these studies, the design of the quantizers is carried out using information from the

plant; namely, these quantizers are based on model-based approach. Thus, if the model of the plant is inaccurate, then the quantizers will be faulty.

Accordingly, in this paper, the data-driven approach is considered for the design of feedback type dynamic quantizers. Besides, this paper presents a class of dynamic quantizers that are constructed using feedforward neural networks. The quantizer, called *neural network quantizers*, are designed using time series data of plant inputs and outputs. Some advantages of this approach are that a model of the plant is not required for the design, i.e., model-free design, and that the quantizer can be designed not only for linear but also for nonlinear plants. The selection of neural networks to perform this job is motivated by the fact that feedforward neural networks are very flexible and that they can be used to represent any nonlinear function/system, in this sense, they work as universal approximators [10,11]. This property is especially important for the design of optimal dynamic quantizers because their structures are functions of the plants' model [4,5]. If the model of the plant is not given, but it is known to be linear, then the structure of the optimal quantizer is also known to be linear. However, if the plant is nonlinear and its model is absent, then the optimal quantizer's structure is unknown. Thus, the neural network can approximate the optimal quantizer's structure based on a series of plant input and output data.

This paper is structured as follows. First, we propose a class of dynamic quantizers composed of feedforward neural network, memories, and a static quantizer. The proposed quantizer has two variations in neural network structures: one is based on a regression-based approach, and the other is based on a classification-based approach. Then, we formulate the quantizers design problem that finds the parameters of the neural network and the quantization interval for given a time series data of plant input and output. Next, with numerical examples, the effectiveness of these quantizers and their design method are verified. Finally, several design variations are considered in order to optimize the quantizer's performance, and comparisons among these variations are carried out.

It should be remarked that various results on the quantizer design for networked control systems have been obtained, e.g., [12–16]. However, the contributions of this paper are distinguished from them as follows. The papers in [12–14] focus on the zoom-in and zoom-out strategy based dynamic quantizer, i.e., the quantizer with time-varying quantization interval. Besides, the paper [15] considers the static logarithmic quantizer, i.e., its quantization interval is not uniform. On the other hand, this paper proposes dynamic quantizer with the time-invariant and uniform interval. Furthermore, the paper [16] proposes a $\Delta\Sigma$ modulator, which is related to the proposed quantizer. Although the result in [16] is restricted to the case of two quantization levels, this paper can deal with the case of multi-levels.

This paper is a journal version of our previous conference papers that were presented in [17,18]. The main difference between this paper and its predecessors is as follows. The system description and problem formulation are improved, and detail explanations of the proposed quantizers are added. Then, we use the ANOVA test to analyze the several simulation results. Besides, we take into account different activation functions for the neural networks hidden layers and compares different initialization methods for the network tuning.

## 2. Neural Network Quantizers

In this section, we first describe a networked control system considered here. Then we present a quantizer, composed of neural networks and a static quantizer, called neural network quantizer.

### 2.1. System Description

This paper considers the system depicted in Figure 1. This system is composed of a plant $P$, a communication channel that has no loses or delays, and the neural network quantizer $Q_{NN}$ proposed in this paper.

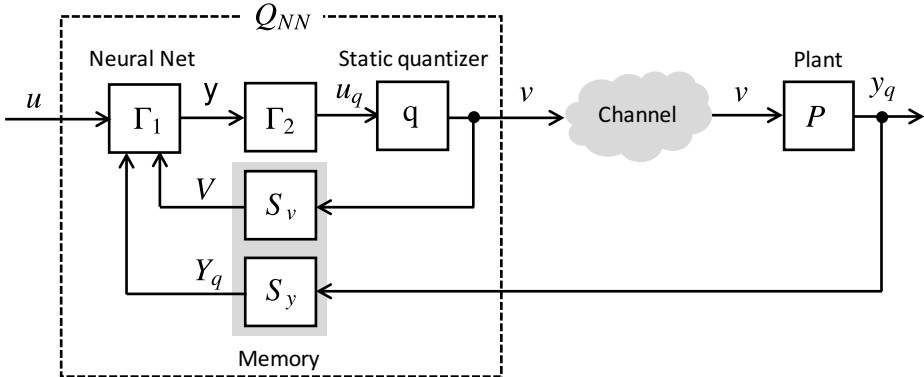

**Figure 1.** Considered system with a neural network quantizer $Q_{NN}$.

The plant is represented by the following single-input-single-output (SISO) model:

$$P : \begin{cases} x(k+1) &= f\left(x(k), v(k)\right), \\ y(k) &= g\left(x(k)\right), \end{cases} \tag{1}$$

where $k \in \{0\} \cup \mathbb{N}$ is the discrete time, $x \in \mathbb{R}^{n_P}$ is the state vector with initial value $x(0) = x_0$, $v \in \mathbb{R}$ is the input, and $y \in \mathbb{R}$ is the output. The functions $f : \mathbb{R}^{n_P} \times \mathbb{R} \to \mathbb{R}^{n_P}$ and $g : \mathbb{R}^{n_P} \to \mathbb{R}$ are in general nonlinear mappings. It is assumed that $f$ and $g$ are continuous and smooth.

The quantizer $Q_{NN}$, shown in Figure 1, is composed of a neural network, a static quantizer q, and a couple of memories $S_v$ and $S_y$. The quantizer is represented are by the following expression:

$$Q_{NN} : \begin{cases} y(k) &= \Gamma_1\left(u(k), V(k), Y_q(k)\right), \\ u_q(k) &= \Gamma_2\left(y(k)\right), \\ v(k) &= q\left(u_q(k)\right), \end{cases} \tag{2}$$

where $u \in \mathbb{R}$ is the input, $y \in \mathbb{R}^{K_{n_L}}$ is the output of the neural network, $u_q \in \mathbb{R}$ is the input of q, and $v \in \{\pm \frac{d}{2}, \pm 2\frac{d}{2}, \cdots, \pm \frac{M}{2} \frac{d}{2}\}$ is the output of q, i.e., the output of $Q_{NN}$. Note that $d$ is the quantization interval, and $M$ is the number of quantization levels which is determined from the data rate of network channel. The signals $V(k)$ and $Y_q(k)$ are the outputs of the memories $S_v$ and $S_y$, respectively. They are time series of past values of the quantized inputs $v(k)$ and the outputs $y_q(k)$ of the plant, and they are given by

$$V(k) = \left[ v(k-1), v(k-2), \dots, v(k-n_V) \right]^\top, \tag{3}$$

$$Y_q(k) = \left[ y_q(k-1), y_q(k-2), \dots, y_q(k-n_Y) \right]^\top, \tag{4}$$

where $n_V$ and $n_Y$ are the dimensions of these memories. Thus, the proposed quantizer is tuned by using the past input and output data of the plant. This means that the quantizer may capture the dynamics of the plant.

This paper proposes two types of neural network quantizers: $Q_{NNR}$ and $Q_{NNC}$. The quantizers $Q_{NNR}$ and $Q_{NNC}$ differ in the expressions of the nonlinear functions $\Gamma_1(\cdot)$, $\Gamma_2(\cdot)$, and $q(\cdot)$. The illustrations of $Q_{NNR}$ and $Q_{NNC}$ are shown in Figure 2. Although detail explanations of two quantizers will be shown in the following subsections, the main difference between them is in the neural network's structure. In $Q_{NNR}$ the network has only one output that shapes the input signal, i.e., the network is trained to perform regression. On the other hand, in $Q_{NNC}$ the network has as many outputs as the considered amount of quantization levels $M$. Each output represents the probability that a given input is matched with a specific quantization level, i.e., the network is trained for classification. Besides, in Figure 2a, the numbers $2, 1, 1, 3, 3$ mean the selected quantization levels, which are determined by the function $\Gamma_2$.

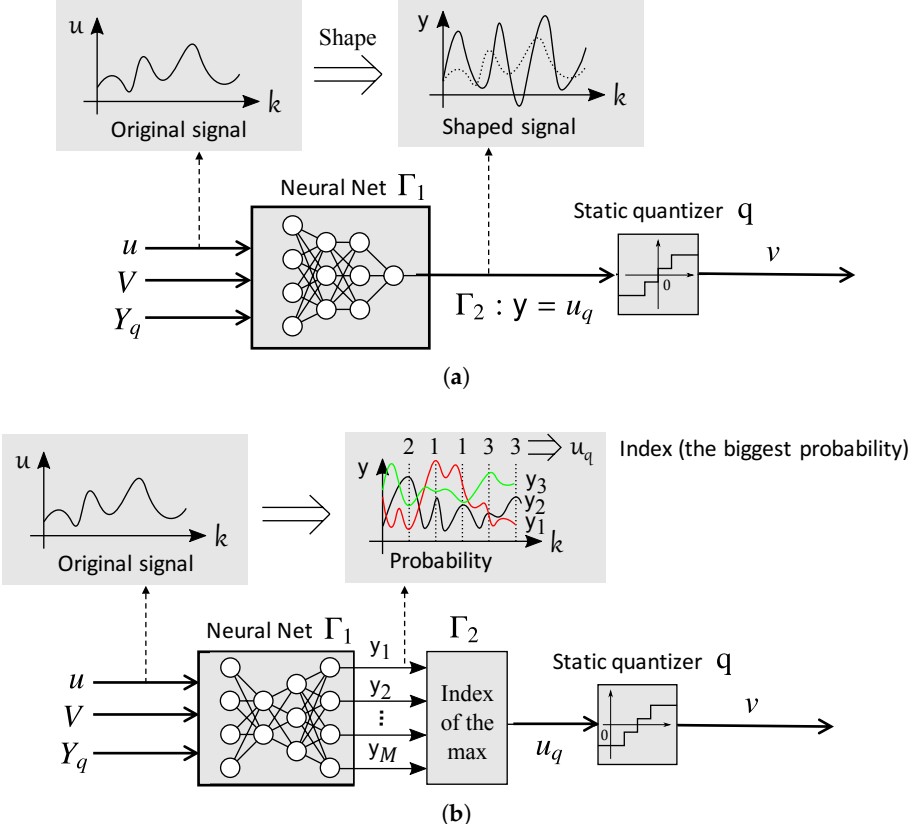

**(a)**

**(b)**

**Figure 2.** Difference between the neural network quantizer based on regression and the one based on classification. (**a**) Regression based approach. The neural net has one output and shapes the original signal; (**b**) Classification based approach. The number of the neural network output is same as that of quantization levels, and each output correponds to the probability that a original signal is classified into a specific quantization level.

## 2.2. Regression Based Neural Network Quantizer

For the regression-based neural network quantizer $Q_{NNR}$, the static quantizer $q(\cdot)$ is a regular finite-level static quantizer with saturation as shown in Figure 3. It receives directly the continuous output of the neural network $u_q(k)$ and rounds it to the nearest discrete value to generate $v(k)$. It has two parameters: one is the number of quantization levels $M \in \mathbb{N}$ and the other is the quantization interval $d \in \mathbb{R}$ with $d > 0$. Figure 3 shows an example of this static quantizer with $M = 4$.

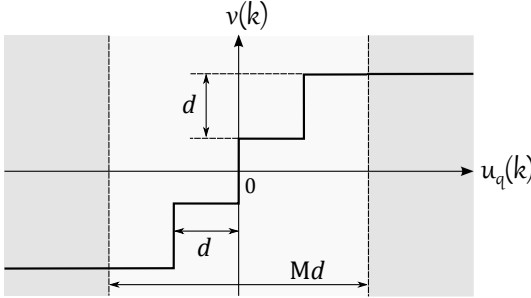

**Figure 3.** Example of a static quantizer $q(\cdot)$ for $Q_{NNR}$ ($M = 4$).

In this paper, the fully connected feed-forward type neural network is adopted to build the function $\Gamma_1(\cdot)$ in Equation (2). An example is shown in Figure 4. In this case, the function $\Gamma_1$ is given by

$$\Gamma_1 : \; y_l = \sum_{k=0}^{K_2} \mathsf{w}_{kl}^{(3)} \mathsf{h} \left( \sum_{j=0}^{K_1} \mathsf{w}_{kj}^{(2)} \mathsf{h} \left( \sum_{i=0}^{K_0} \mathsf{w}_{ji}^{(1)} \mathsf{x}_i \right) \right) \quad \text{for } l = 1, 2, \ldots, K_{n_L}. \tag{5}$$

The following elements can be recognized in the network, the input units $\mathbf{x} \in \mathbb{R}^{K_0}$, the output units $\mathbf{y} \in \mathbb{R}^{K_{n_L}}$, and the hidden units $\mathbf{z}^{(i)} \in \mathbb{R}^{K_i}$ ($i = 1, 2, \ldots, n_L - 1$), where $n_L$ is the number of layers in the network. Note that the inputs of this network are $u(k)$, $V(k)$ and $Y_q(k)$, and they are represented as $\mathbf{x}(k) = [u(k), \, V^\top(k), \, Y_q^{\;\top}(k)]^\top$. Besides, the size of the neural network is represented by $\mathbf{K} = [K_1, K_2, \ldots, K_{n_L}]$. Each neuron performs a nonlinear transformation of a weighted summation of the previous layer outputs as follows

$$\mathsf{z}_j^{(l)} = \mathsf{h} \left( \sum_{i=0}^{K_l} \mathsf{w}_{ji}^{(l)} \mathsf{z}_i^{(l-1)} \right), \tag{6}$$

where $\mathsf{w}_{ji}^{(l)}$ represents the weight of the connection that goes from the $i$th neuron in layer $(l-1)$ to the $j$th neuron in layer $l$. Notice here that a simplified notation is used, where instead of having biases the units $\mathsf{x}_0 = 1$ and $\mathsf{z}_0^{(l-1)} = 1$ are included in the network. Then, because these elements are constants, their respective connection's weight $\mathsf{w}_{j0}^{(l)}$ serves as bias parameters. The weights of all the connections in the network are put together in a vector $\mathbf{w}$ called the *weights vector* that has dimension $n_{\mathsf{w}} = \sum_{i=0}^{n_L - 1} (K_i + 1) K_{i+1}$. Furthermore, $\mathsf{h}(\cdot)$ represents the nonlinear transformation and is called *activation function*. There are many functions that serve as activation functions such as logistic sigmoid, hyperbolic tangent, and rectified linear unit (ReLU). In this paper, we adopt the most commonly used sigmoid function:

$$\text{sigm}(a) \quad = \quad \frac{1}{1 + \exp{(-a)}}. \tag{7}$$

Finally, since $K_{n_L} = 1$ in $Q_{NNR}$, the function $\Gamma_2$ is given by $\Gamma_2 : u_q = y_l$.

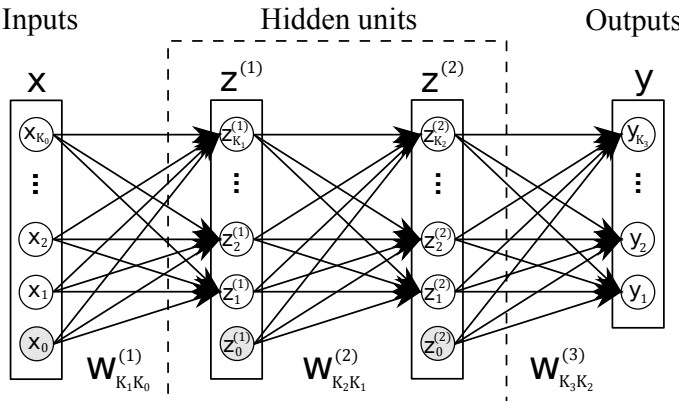

**Figure 4.** Fully connected 3 layered feed-forward neural network example.

### 2.3. Classification Based Neural Network Quantizer

For the classification based neural network quantizer $Q_{NNC}$, the static quantizer $\mathsf{q}(\cdot)$ is not a conventional one. Its input $u_q(k)$ comes from a set of indexes, each of which makes reference to a specific quantization level, i.e., $u_q(k) \in \{1, 2, \ldots, M\}$. Thus, $\mathsf{q}(\cdot)$ is adapted to match each index to the corresponding quantization level as Figure 5 shows. This quantizer is also defined by the number of quantization levels $M$ and the quantization interval $d$.

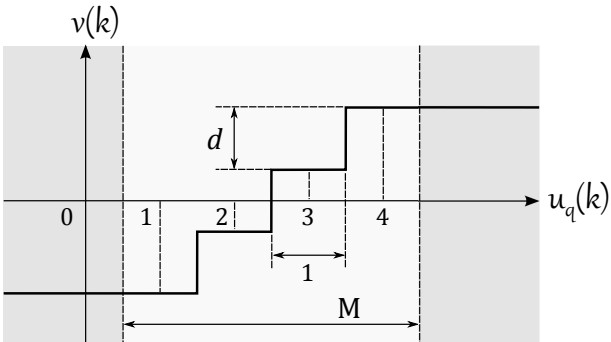

**Figure 5.** Example of static quantizer q[·] adapted for $Q_{NNC}$ ($M = 4$).

The neural network $\Gamma_1$ in $Q_{NNC}$ is same as that in $Q_{NNR}$. The inputs of this network are the same as in the previous case $\mathbf{x}(k) = [u(k), \mathbf{V}^\top(k), \mathbf{Y_q}^\top(k)]^\top$ and the hidden units activation function $\mathsf{h}(\cdot)$ is also the logistic sigmoid in Equation (7). The dimension of the ouput is $K_{n_L} = M$. Then each output of the network $\mathsf{y}_i(k)$ is associated with one quantization level, and represents the probability that a given input is classified into a specific quantization level. Therefore, the quantization level with the biggest probability is selected to be the network's output, and it is given by

$$\Gamma_2: \; u_q(k) = \underset{i \in \{1,2,\dots,M\}}{\arg\max} \; \frac{\exp\left(\mathsf{y}_i(k)\right)}{\sum_{j=1}^{K_{n_L}} \exp\left(\mathsf{y}_j(k)\right)}. \tag{8}$$

## 3. Quantizer Design Problem

In this paper, it is assumed that the number of quantization levels $M$, the memory sizes $n_V$ and $n_Y$, and the neural network structure $K$ are given. Thus, the design parameters are the weight vector $\mathbf{w}$ and the quantization interval $d$ of the neural network quantizer $Q_{NN}$.

The performance of the quantizer $Q_{NN}$ in NCSs can be evaluated using a construction known as *error system*. The considered error system is depicted in Figure 6. This system is composed of two branches. In the lower branch, the input signal $u$ is applied directly to the plant $P$ that produces the ideal output $y$. In the upper branch the effects of quantization are considered and $u$ is applied to the quantizer $Q_{NN}$ that generates the quantized signal $v$ that is applied to the plant. The output of the plant in this case is represented by $y_q$, and the difference $y_q - y$ is the error signal. The error signal $e(k)$ is used to evaluate the performance degradation of the system. By minimizing $y_q - y$, the system composed of the quantizer $Q_{NN}$ and the plant $P$ can be optimally approximated to the plant $P$, in terms of the input-output relation.

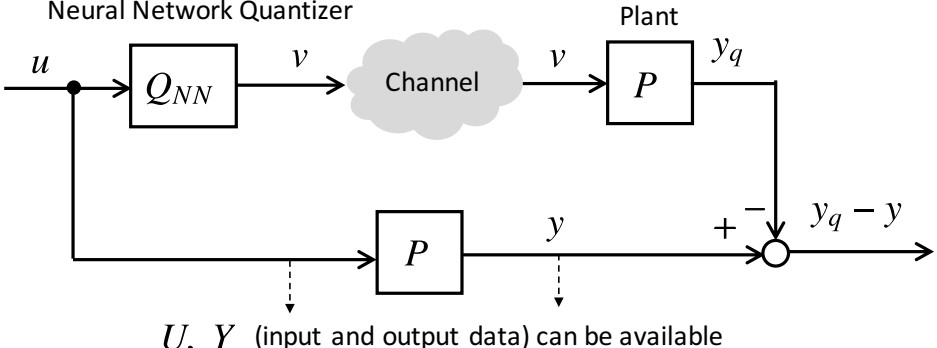

**Figure 6.** Error system.

In this context, a parameter known as *performance index* is used to measure the system's performance degradation. The performance index considered here is the sum-of-squares error function that is defined by

$$E(Q_{NN}) = \sum_{k=0}^{n_s} \left[ y_q(u(k), \mathbf{w}, d) - y(k) \right]^2 \tag{9}$$

where $u(k)$ is used to build $\mathbf{x}(k)$ along side with $V(k)$ and $Y_q(k)$ that are generated dynamically. It is necessary to make $E(Q_{NN})$ as small as possible to maintain the output error low. Then, the design of $Q_{NN}$ is set up as an optimization problem in which the performance index is minimized.

This paper assumes that, although the model is unknown, it is possible to feed it with inputs and measure its outputs. Then, a time series of inputs and outputs of the plant will be available. These time series are represented as follows.

$$\mathbf{U} = [u(1), u(2), \dots, u(n_s)]^\top, \tag{10}$$

$$\mathbf{Y} = [y(1), y(2), \dots, y(n_s)]^\top, \tag{11}$$

where $n_s$ is the length of the time series, namely, the number of samples. Notice that $y(k)$ ($k = 1, 2, \dots, n_s$) represents the output of the plant $P$ when $u(k)$ is applied directly to it, i.e., $v(k) = u(k)$.

Then, the neural network quantizers design problem is formulated as follows:

**Problem 1.** *Suppose that the time series data $\mathbf{U}$ and $\mathbf{Y}$ of the plant, the number of quantization levels $M$, the neural network structure $K$, and the memory sizes $n_V$ and $n_Y$ are given. Then, find the parameters of $Q_{NN}$: the weight vector $\mathbf{w}$ and the quantization interval d which minimize $E(Q_{NN})$, under the condition that $d > 0$.*

This design problem is nonlinear and nonconvex. Thus, it cannot be solved using gradient-based optimization methods such as linear programming or quadratic programming. Moreover, conventional neural network training techniques based on error backpropagation cannot be used either due to the structure of the system, as it was explained previously. Therefore, alternative optimization methods should be used.

In this regard, the metaheuristics stand out from the available options because of their flexibility and a wide variety of implementations [19]. In particular, the differential evolution (DE) metaheuristic algorithm is used to perform the design of $Q_{NN}$. This choice is justified by the fact that DE has proven to be effective in the training of neural networks [20,21] and that it has shown an outstanding performance in the design of dynamic quantizers [9]. DE is a population-based metaheuristic algorithm inspired in the mechanism of biological evolution [22,23]. In this algorithm, the cost function $J(\boldsymbol{\theta})$ is evaluated iteratively over a population of possible solutions or *individuals* $\boldsymbol{\theta}_i \in \mathbb{R}^n$ in each iteration the individuals improve their values and move towards the best solution. Finally, the individual with the lowest fitness value in the last iteration is regarded as the optimal solution. Some advantages of DE are that it is very easy to implement and has only two tuning parameters: the scale factor $F$ and the crossover constant $H$, apart from the number of individuals $N$ and the maximum number of iterations $t_{max}$. Besides, DE shows very good exploration capacities and converges fast to global optima. DE has many versions and variations; the one considered in this study is the classical DE/best/1/bin strategy, which is described in Algorithm 1.

---

**Algorithm 1 : DE** (DE/best/1/bin strategy)

---

**Initialization:** Given $N \in \mathbb{N}$, $t_{max} \in \mathbb{N}$, $F \in [0,2]$, $H \in [0,1]$ and the initial search space $S_0 = [\theta_{min}, \theta_{max}]^n$. Set $t = 0$ then select randomly $N$ individuals $\{\theta_1, \theta_2, \ldots, \theta_N\}$ in the search space.

**Step 1:** The cost function $J(\theta)$ is evaluated for each $\theta_i$ and $\theta_{base} = \theta_l$ is calculated by:

$$l = \underset{i \in \{1,2,\ldots,N\}}{\arg\min} \ J(\theta_i). \tag{12}$$

If $t = t_{max}$ then $\theta_{base}$ is the final solution, if not go to **Step 2**.

**Step 2 (Mutation):** For each $\theta_i$ a mutant vector $\mathcal{M}_i$ is generated by:

$$\mathcal{M}_i = \theta_{base} + F(\theta_{\tau_{1,i}} - \theta_{\tau_{2,i}}), \tag{13}$$

where $\tau_{1,i}$ and $\tau_{2,i}$ are random indexes subject to $i \neq \tau_{1,i} \neq \tau_{2,i} \neq l$.

**Step 3 (Crossover):** For each $\theta_i$ and $\mathcal{M}_i$ a trial vector $\mathcal{T}_i$ is generated by:

$$\mathcal{T}_{i,j} = \begin{cases} \mathcal{M}_{i,j} & \text{if } \rho_{i,j} \leq H \text{ or } j = j_{rand}, \\ \theta_{i,j} & \text{otherwise}, \end{cases} \tag{14}$$

where $\rho_{i,j} \in [0,1]$ and $j_{rand} \in \{1, 2, \ldots, n\}$ are generated randomly.

**Step 4 (Selection):** The members of the next generation $k + 1$ are selected by:

$$\theta_i \leftarrow \begin{cases} \mathcal{T}_i & \text{if } J(\mathcal{T}_i) \leq J(\theta_i), \\ \theta_i & \text{otherwise}, \end{cases} \tag{15}$$

then $t \leftarrow t + 1$ and go to **Step 1**.

---

Since the design parameters of $Q_{NN}$ are **w** and $d$, an individual for the DE algorithm will have the following form $\theta = [d \ \ \mathbf{w}]^\top$ with dimension $n = 1 + n_{\mathbf{w}}$. From these parameters, the weights vector **w** is not affected by any constraint, but the quantization interval $d$ should always be positive $d > 0$. DE has no direct way to handle the constraints of the optimization problem since it was designed to solve unconstrained problems. Then, in order to manage the constraint condition, a method developed by Maruta et al. in [24] is employed. This method transforms the constrained optimization problem into the following unconstrained one.

$$\underset{\theta \in \mathbb{R}^n}{\text{minimize}} \, J(\theta) \quad \text{for} \quad J(\theta) := \begin{cases} \arctan[E(\theta)] - \pi/2 & \text{if } d > 0, \\ -d & \text{otherwise}, \end{cases} \tag{16}$$

where $E(\theta)$ is the performance index in Equation (9). This constraints management method ensures that $d$ is positive.

The learning resulting from the training of a deep neural network depends highly on the initial weights of the network because many of the learning techniques are in essence local searches. Therefore, it is very important to initialize the network's weights appropriately [25,26]. There are several ways to initialize the neural networks to perform the training. The most common method is the uniformly random initialization where random values sampled from a certain interval using a uniform probability distribution are assigned to the weights and biases of the network. The initialization intervals are selected according to, but they are usually small and close to zero. Popular ones are the intervals $[-1, 1]$ or $[-0.5, 0.5]$. Another prevalent type of initialization was developed in [27] by Glorot and Bengio. This method is known as Xavier Uniform initialization (from Xavier Glorot). In this method, the weights of each layer in the network are initialized using random uniform sampling in a specific interval

$$\mathbf{w}_i \sim U[-l_i, \ \ l_i] \quad \text{for} \quad i = 1, 2, \ldots, n_L, \tag{17}$$

where $\mathbf{w}_i$ represents the weights of the $i$th layer. The limits of the interval are given by $l_i$ which is a function of the number of neurons of the considered layer $K_i$, the number of neurons in the previous layer $K_{i-1}$ and the hidden layers activation function h. The limit is the following

$$l_i = \frac{4\sqrt{6}}{\sqrt{K_{i-1} + K_i}} \quad \text{for} \quad h(a) = \text{sigm}(a). \tag{18}$$

## 4. Numerical Simulations

To verify that the proposed neural network quantizers and their design method work properly, several numerical simulations were performed. In these simulations, the following discrete, nonlinear and stable plant is used.

$$P: \begin{cases} \begin{bmatrix} x_1(k+1) \\ x_2(k+1) \end{bmatrix} = \begin{bmatrix} f_1(\mathbf{x}(k)) + f_3(\mathbf{x}(k))u(k) \\ f_2(\mathbf{x}(k)) + f_3(\mathbf{x}(k))u(k) \end{bmatrix}, \\ \\ y(t) = 1.45x_1(k) + x_2(k), \end{cases} \tag{19}$$

$$\begin{aligned} f_1(\mathbf{x}) &= 0.8x_1 - 0.4x_2 + 0.4e^{-|x_2|}\cos^3(x_1), \\ f_2(\mathbf{x}) &= 0.6x_2 + 0.4e^{-|x_1|}|\cos(x_1)|^{\frac{1}{2}}, \\ f_3(\mathbf{x}) &= 0.01 + 0.01((x_1)^4 + 0.1)^{-1}. \end{aligned}$$

This plant is a modified version of the plant shown in [5]. The initial state is $x_0 = [0.1, \ -0.2]^\top$, and the input signal used in the examples is given by

$$u(k) = 0.3\sin(6k) + 0.4\sin(k) + 0.3\sin(3k). \tag{20}$$

The evaluation interval is $L = 1000$, which implies that the amount of samples taken is $n_s = 1000$.

The quantizers are constructed with $n_Y = n_V = 5$, $M = \{2, \ 8\}$ and neural networks with $n_L = \{2, \ 4\}$. Given the size of the memories and the dimension of $u(k)$ all the networks have inputs with dimension $K_0 = 11$. The neural networks' structure depends on the type of quantizer and $M$. Table 1 summaries the structure of the quantizers used in the simulations. For the regression case (R) the network's structure and the dimension of $\mathbf{w}$ ($n_w$) are independent of $M$. This is not the case for the classification type of quantizer (C). Table 1 also shows a comparison among the $n_w$ of each network.

**Table 1.** Simulation conditions.

| Quantizer Type | $M$ | $K$ | $n_L$ | $n_w$ | $n$ |
|---|---|---|---|---|---|
| R: $Q_{NNR}$ | {2, 8} | [10, 1] | 2 | 132 | 133 |
| | | [10, 10, 10, 1] | 4 | 352 | 353 |
| C: $Q_{NNC}$ | 2 | [10, 2] | 2 | 142 | 143 |
| | | [10, 10, 10, 2] | 4 | 362 | 363 |
| | 8 | [10, 8] | 2 | 208 | 209 |
| | | [10, 10, 10, 8] | 4 | 428 | 429 |

The hyper parameters of DE are $N = 500$, $t_{max} = 2000$, $F = 0.6$, and $H = 0.9$. The simulations were performed $N_{run} = 50$ times for each considered case. Then, since the individuals have the form $\boldsymbol{\theta} = [d \ \mathbf{w}]$ the dimensions of the optimization problems $n$ will be the ones shown is the last column of Table 1. Looking at Table 1 it is possible to see that $Q_{NNC}$ has more parameters than $Q_{NNR}$, this is a factor that influences the performance of the proposed design method.

The DE individuals are initialized as follows. The first element $d$ is uniformly sampled from the interval $(0, 1]$. The network weights are initialized using the uniform random and the Xavier uniform initialization methods, described in Section 3.

After running the DE algorithm $N_{run} = 50$ times for each considered case, the quantizers $Q_{NN}$ with the lowest $E(\mathbf{w}, d)$ are selected to be the optimal quantizers. Then, in order to test these quantizers, the error system in Figure 6 is fed with the input signal $u(k)$ for each case. It results that all the quantizers work properly and show good performance. For instance, Figure 7 depicts the signals resulting from applying $u(k)$ to the system with the quantizers designed for $M = 2$ and $n_L = 2$. This figure shows that the output signals obtained by quantization $y_q(k)$ follow the ideal output signal $y(k)$ pretty well and that the error between them is small in both cases. Also, the inputs $u_q(k)$ of the static quantizers are shown for comparison.

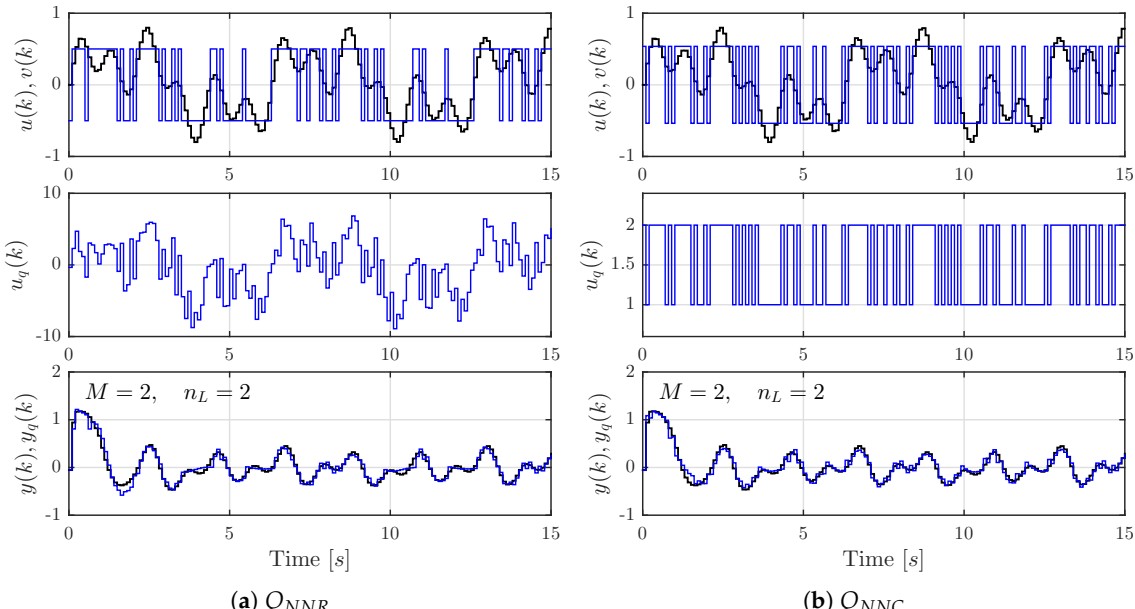

(a) $Q_{NNR}$        (b) $Q_{NNC}$

**Figure 7.** Signals resulting from applying $u(k)$ to the system with the $Q_{NN}$ designed for $M = 2$ and $n_L = 2$. The black lines represent the signals without quantization ($u(k)$, $y(k)$) and the blues ones are the signals when quantization is applied ($v(k)$, $u_q(k)$, $y_q(k)$).

To further validate this observation, in Figure 8 there are shown the output signals of the system where the neural network quantizers were designed for $M = 2$ and $n_L = 4$, and in Figure 9 the ones for $M = 8$, $n_L = 2$ and $n_L = 4$. From these, we see that the proposed quantizer works well.

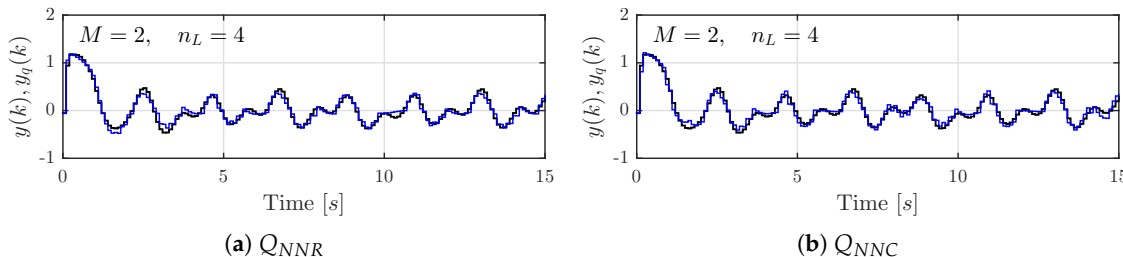

(a) $Q_{NNR}$        (b) $Q_{NNC}$

**Figure 8.** Output signals $y_q(k)$ (blue) and $y(k)$ (black) resulting from applying $u(k)$ to the error system with $Q_{NN}$ designed for $M = 2$ and $n_L = 4$.

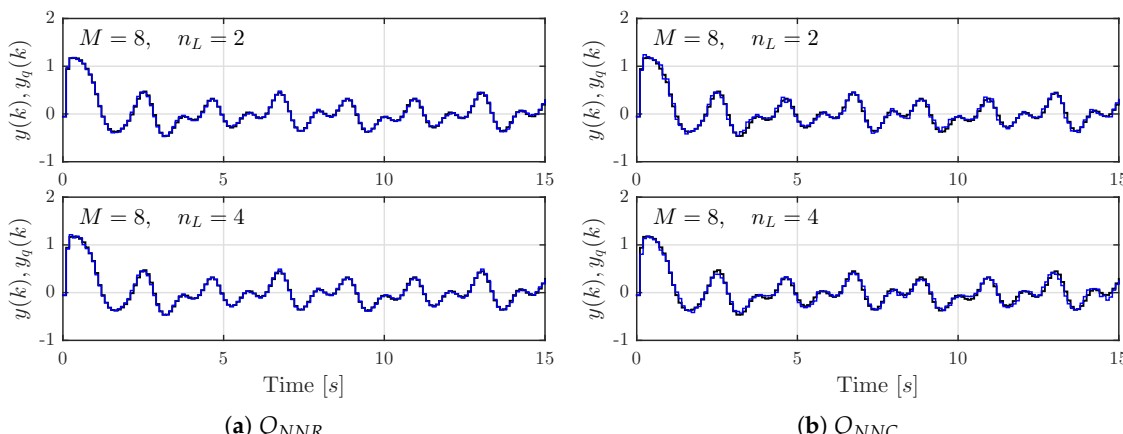

**Figure 9.** Output signals $y_q(k)$ (blue) and $y(k)$ (black) resulting from applying $u(k)$ to the error system with $Q_{NN}$ designed for $M = 8$, $n_L = 2$ (**upper figure**) and $n_L = 4$ (**lower figure**).

In addition, the result with the static quantizer q case and the result with the optimal dynamic quantizer case proposed in [5] are shown in Figure 10 for comparison. The value of the performance index for the static quantizer is $E(q) = 7.9961$ and that for the optimal dynamic quantizer is $E(Q_{NNR}) = 3.6002$. On the other hand, the performance of the proposed quantizer $Q_{NNR}$ for $M = 2$ and $n_L = 2$ is $E(Q_{NNR}) = 3.73724$ and that for $M = 2$ and $n_L = 4$ is $E(Q_{NNR}) = 3.66764$. From this comparison result, we see that the proposed quantizer achieves higher performance than the static quantizer. Then, we find that the proposed quantizer is similar to the optimal dynamic quantizer, although the proposed quantizer is designed with the time series data of plant inputs and outputs, i.e., without the model information of the plant. Therefore, we can confirm that the neural network in the proposed quantizer captures the dynamics of the plant appropriately based on the time series data of the plant input and output.

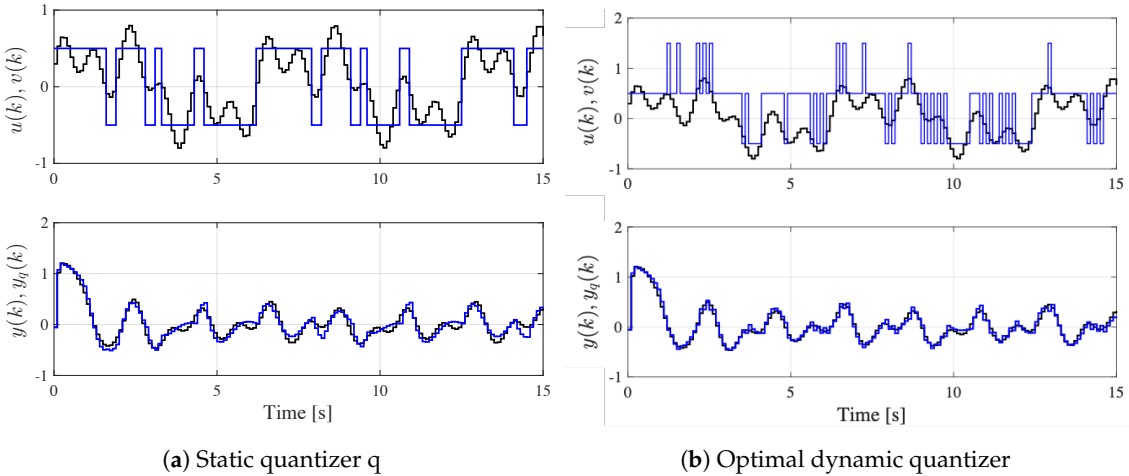

**Figure 10.** Signals resulting from applying $u(k)$ to the systems with the static quantizer q in Figure 3 and the optimal dynamic quantizer proposed in [5]. The black lines represent the signals without quantization ($u(k)$, $y(k)$) and the blues ones are the signals when quantization is applied ($v(k)$, $u_q(k)$, $y_q(k)$).

The minimum values of the performance indexes in Equation (9), found by DE, are listed in Table 2. In addition, this table lists the average performance indexes and their standard deviation. The values in this table are divided according to their $M$, initialization method, $n_L$ and type (regression or classification). There are two initialization methods implemented: uniform random (Urand) and Xavier.

**Table 2.** $E(Q_{NN})$ analysis for h = sigmoid ($N_{run} = 50$).

| M | Init. | $n_L$ | Type | Min. | Avg. | Std. Dev. |
|---|---|---|---|---|---|---|
| 2 | Urand | 2 | R | 3.73724 | 4.32987 | 0.48300 |
| | | | C | 3.66946 | 4.27038 | 0.34762 |
| | | 4 | R | 3.66764 | 4.22516 | 0.45819 |
| | | | C | 3.54773 | 4.30158 | 0.49296 |
| | Xavier | 2 | R | 3.42879 | 4.15830 | 0.36054 |
| | | | C | 3.53307 | 4.09696 | 0.35038 |
| | | 4 | R | 3.65081 | 4.10635 | 0.35909 |
| | | | C | 3.63822 | 4.04066 | 0.29597 |
| 8 | Urand | 2 | R | 0.17825 | 0.20622 | 0.01612 |
| | | | C | 0.91201 | 2.91111 | 1.54333 |
| | | 4 | R | 0.22911 | 0.32243 | 0.14041 |
| | | | C | 0.81667 | 2.81630 | 1.43382 |
| | Xavier | 2 | R | 0.20424 | 1.90045 | 1.22825 |
| | | | C | 0.93284 | 2.85580 | 1.44467 |
| | | 4 | R | 0.24762 | 0.85188 | 1.03043 |
| | | | C | 1.00016 | 2.61201 | 1.06714 |

Drawing conclusions from this table by simple observation is difficult. For example, looking at the minimum values of $E(Q_{NN})$ in the case of $M = 2$, it is possible to say that $Q_{NNC}$ have better performance (smaller $E(Q_{NN})$) than $Q_{NNR}$ in most cases. The average values not always corroborate this observation. For $M = 8$, $Q_{NNR}$ has the smallest value of $E(Q_{NN})$ in each case. However, there is no evidence that there is a significant difference in the performance of these types of quantizers. Therefore, the analysis of variance (ANOVA) is used to check if there are significant differences or not among these values.

Because many factors influence $E(Q_{NN})$, the 3-way ANOVA (ANOVA with three factors) is used. The considered factors are Type, initialization method (Init.) and number of layers $n_L$. The categories of each factor are known as elements. For instance, the elements of the factor Type are R ($Q_{NNR}$: regression) and C ($Q_{NNC}$: classification). The $M$ is not taken as a factor, because $M = 8$ gives smaller $E(Q_{NN})$s than $M = 2$. Then, it is not necessary to check which one gives better results. The considered significance level is $\alpha = 0.05$. The goal is to determine if there is some statistical difference among the $E(Q_{NN})$'s means of the design methods.

A summary of this test is shown in Table 3. The ANOVA test shows if there are significant differences among sets of data. When doing 3-way ANOVA, it is possible to see not only if there is a significant difference among elements of a factor but also combinations of elements of different factors. In this particular case, it will tell if there is a significant difference between $Q_{NNR}$ and $Q_{NNC}$, and also it will tell if there are differences among the combinations of the quantizer types and the initialization methods. Then, the 3-way ANOVA test is run separately for $M = 2$ and $M = 8$. For the case of $M = 2$, the significant difference is found only for the initialization method. For the case of $M = 8$ the significant difference is found for all the factors and the combinations of them with the exception of the combination of the quantizer type and $n_L$.

**Table 3.** Tukey pairwise comparison for h = sigmoid and single factors]Tukey pairwise comparison 3-way ANOVA for h = sigmoid and single factors. Grouping information using the Tukey test and 95% confidence. *Means that do not share a letter are significantly different.*

| *M* | Factor | | N | Mean | Grouping | |
|---|---|---|---|---|---|---|
| | Type | R | 200 | 4.20492 | A | |
| | | C | 200 | 4.17739 | A | |
| 2 | Init | Urand | 200 | 4.28175 | A | |
| | | Xavier | 200 | 4.10057 | | B |
| | $n_L$ | L2 | 200 | 4.21388 | A | |
| | | L4 | 200 | 4.16844 | A | |
| | Type | C | 200 | 2.79881 | A | |
| | | R | 200 | 0.82024 | | B |
| 8 | Init | Xavier | 200 | 2.05504 | A | |
| | | Urand | 200 | 1.56401 | | B |
| | $n_L$ | L2 | 200 | 1.96839 | A | |
| | | L4 | 200 | 1.65066 | | B |

So far only one type of activation function $h(\cdot)$, the sigmoid function, have been used in the hidden layers to build the neural networks. However, there are other activation functions that can be used. In this section two additional activation functions are considered: the hyperbolic tangent (tanh) and the Rectified Linear Unit (ReLU). These functions were defined by

$$\tanh(a) = \frac{1 - \exp(-2a)}{1 + \exp(-2a)}, \tag{21}$$

$$\text{ReLU}(a) = \max(a, 0). \tag{22}$$

In addition, the limits $l_i$ in (17) for Xavier Uniform initializatin are given by

$$l_i = \frac{\sqrt{6}}{\sqrt{K_{i-1} + K_i}} \quad \text{for} \quad h(a) = \tanh(a), \tag{23}$$

$$l_i = \frac{\sqrt{6}}{\sqrt{K_i}} \quad \text{for} \quad h(a) = \text{ReLU}(a). \tag{24}$$

Several numerical simulations were performed to compare the performance of the neural network quantizers built with these functions. The settings of these simulations are the same as in the previous cases where h = sigm, but they were carried out only for $M = 8$. These simulations were run $N_{run} = 50$ times for each case. The results are summaries in Table 4.

As before, it is difficult to conclude from the table by simple observation. Therefore, the ANOVA test is used to analyze the data. In this case, four factors influence the results: h, initialization method, $n_L$ and quantizer type. However, because the influence of $n_L$ is understood the focus in this section will be in the factors: h, initialization method, quantizer type, and the interaction among each other. Therefore, the 3-way ANOVA general linear model of $E(Q_{NN})$ versus quantizer type (Type), initialization method (Init) and activation function h is considered. The significance level used in this analysis is $\alpha = 0.05$. The analysis of variance showed that the statistical null hypothesis that all the means are the same was rejected for every single factor and the combination of them. This means that in each case there is at least one element that significantly differs from the others. The Tukey pairwise comparison is made to see the differences among the quantizer's design elements.

**Table 4.** $E(Q_{NN})$ results summary for h = tanh and h = ReLU ($M = 8$).

| h | Init. | $n_L$ | Type | Min. | Avg. | Std. Dev. |
|---|---|---|---|---|---|---|
| tanh | Urand | 2 | R | 0.17945 | 0.57343 | 0.73814 |
| | | | C | 0.85707 | 2.17040 | 1.21598 |
| | | 4 | R | 0.24340 | 1.83621 | 1.37026 |
| | | | C | 0.69604 | 1.62132 | 0.68768 |
| | Xavier | 2 | R | 0.17010 | 0.20761 | 0.02103 |
| | | | C | 0.92943 | 1.97048 | 0.78436 |
| | | 4 | R | 0.19868 | 0.25325 | 0.03900 |
| | | | C | 0.66041 | 1.77138 | 0.99609 |
| ReLU | Urand | 2 | R | 0.16532 | 0.75854 | 1.21252 |
| | | | C | 0.70494 | 1.93226 | 0.98668 |
| | | 4 | R | 0.19175 | 3.04153 | 1.69148 |
| | | | C | 0.71718 | 1.93823 | 1.27409 |
| | Xavier | 2 | R | 0.22438 | 2.97202 | 1.24661 |
| | | | C | 0.72398 | 2.12915 | 1.12578 |
| | | 4 | R | 0.22062 | 3.01670 | 1.70763 |
| | | | C | 0.63188 | 2.14319 | 1.03974 |

The results of this test are summarized in Table 5. From the results, we see the following things. First, they tell that there is a significant difference between the $Q_{NNR}$ and $Q_{NNC}$, and that $Q_{NNR}$ outperforms $Q_{NNC}$. Second, they show that there is a difference between the initialization methods and that the Urand method exhibits better performance than the Xavier method. These results corroborate the ones shown in Table 3 previously obtained for $M = 8$ and h = sigm. Third, the table shows that the performances of the considered activation functions vary significantly, that the one with the best performance is h = tanh, and that the one with the lowest performance is h = ReLU.

**Table 5.** Tukey pairwise comparison 3-way ANOVA for the activation functions comparison ($M = 8$). Grouping information using the Tukey test and 95% confidence. *Means that do not share a letter are significantly different.*

| | Factor | N | Mean | Grouping | |
|---|---|---|---|---|---|
| Type | C | 600 | 2.23930 | A | |
| | R | 600 | 1.32836 | | B |
| Init | Xavier | 600 | 1.89033 | A | |
| | Urand | 600 | 1.67733 | | B |
| h | ReLU | 400 | 2.24145 | A | |
| | sigm | 400 | 1.80953 | | B |
| | tanh | 400 | 1.30051 | | C |

## 5. Conclusions

This paper introduces the concept of neural network quantizers that are designed using a set of the inputs and outputs of the plant. These quantizers are aimed at systems in which the model of the plant is unknown or unreliable. They are constructed using feedforward neural networks and static quantizers. Two types of neural network quantizers are proposed: regression-based type $Q_{NNR}$ and classification based type $Q_{NNC}$. In addition, a design method based on differential evolution is proposed for these quantizers.

By means of several numerical examples, it was found that both types of neural network quantizers are effective alongside with their DE based design method. Furthermore, many variations were considered in the construction of these quantizers. These variations are reflected in the number

of quantization levels ($M = \{2, 8\}$), in the number of layers of the network ($n_L = \{2, 4\}$), in the type of network initialization technique (Urand, Xavier), and in the hidden layers' activation functions (sigm, tanh, ReLU). Several conclusions were reached based on the analysis of variance performed on the simulations results. Some of the most important is that the quantizers based on regression outperform the ones based on classification, that the best initialization method is the random uniform (Urand), and that the activation function that gives the best performance is tanh.

**Author Contributions:** Conceptualization, Y.M. and J.E.R.R.; methodology, Y.M. and J.E.R.R.; software, J.E.R.R.; validation, J.E.R.R.; formal analysis, J.E.R.R.; investigation, J.E.R.R.; data curation, J.E.R.R.; writing—original draft preparation, J.E.R.R.; writing—review and editing, J.E.R.R. and Y.M.; visualization, J.E.R.R. and Y.M.; project administration, Y.M.; funding acquisition, Y.M.

**Funding:** This research was partly supported by JSPS KAKENHI (16H06094) and a research grant from The Mazda Foundation.

**Acknowledgments:** The authors would like to thank Professor Kenji Sugimoto, Nara Institute of Science and Technology for his support.

**Conflicts of Interest:** The authors declare no conflict of interest.

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
