# Peer review of "Design of Neural Network Quantizers for Networked Control Systems"

_electronics, doi:10.3390/electronics8030318_

Reviewer 1 Report

The paper proposes using neural networks to quantize signals used in a plant control process.

Major comments:

* The authors do not compare their approach against some kind of a baseline (e.g. a static quantizer without the neural network). They only against a signal that is not quantized. It is not clear what would the results be if a simpler quantization method was used. If the results are similarly good, the need for the system proposed in the paper is not apparent. The authors should address this.

Minor comments:
* Figure 2 - the depiction of the classification based approach is difficult to understand, especially without reading the Section 2.3 first. For example, you should make clear that "2 1 1 3 3" are the selected quantization levels.

* I wonder what are the a-priori benefits of the classification vs. regression method? (Do you have any motivation why investigate the classification approach at all, since it is more complex and less intuitive to grasp? Also, in the results classification shows higher errors than regression.)
Perhaps the classification approach might allow quantization where the difference in quantizations levels is not a constant, but varies. The authors don't seem to consider this option, as the quantization interval `d` is a single, scalar value in both methods.

* Any rationale on why did you choose equations (19) and (20)?

* Line 65 has a typo: looses -> loses

Reviewer 2 Report

This paper proposed two types of dynamic quantizers for networked control systems using neural networks and memories: regression based and classification based. Simulation experiments compared their performance and that of their variations to evaluate the effectiveness of the proposed method. Overall, the manuscript is well written in terms of language. However, the following major issues should be carefully addressed.

Content:

In the line 7 of Abstract, the authors emphasized the proposed method is based on a model-free approach. However, explanation and discussion of the model-free approach are not detailed. 

In the 2nd paragraph of the Introduction section, only a few related works [4-9] were introduced, and most of them are from the authors’ group. I suggest the authors introduce more related and recent research works from other research groups and summarized the advantages and disadvantages of the related works. 

Most importantly, the authors claimed that “This paper is a journal version of our previous conference papers that were presented in [2] and [3]”.  However, I found this paper is nearly a combined version of the two conference papers, instead of a substantial extension to the conference papers. The content and the simulation experiments in current form are not substantially extended and enhanced.  

Figure and Table Positions:

Figure positions should be adjusted as they appear near its description in the flow of the text. For example., Fig.3 and Fig.4 are located in page 5 while their description is in page 3. 

Tables 2 and 3 should be moved before References instead of interweaving with the reference section. 

Experiments:

In the simulation, only the two types of proposed dynamic quantizers were compared, as well as their variations. The evaluation experiments are not sufficient. The authors should compare the proposed methods with the state-of-the-art model-free approaches or dynamic quantizers proposed in recent years (e.g. year 2018). 

In addition, Fig. 7 can be improved by combining the curves of two results into one figure instead of separate two subfigures. Fig.8 and Fig.9 have the same issue.

Typos: 

Line 32: are -> were

Line 74: captures -> capture

Fig2 (a) caption: Regresion -> Regression 

Author Response

Round  2

Reviewer 1 Report

The authors have addressed my comments.

Reviewer 2 Report

The authors have addressed the major issues in the revised manuscript. I recommend accepting this manuscript in present form.